# Synthesis of Homodrimane Sesquiterpenoids Bearing 1,3-Benzothiazole Unit and Their Antimicrobial Activity Evaluation

**DOI:** 10.3390/molecules27165082

**Published:** 2022-08-10

**Authors:** Lidia Lungu, Caleria Cucicova, Svetlana Blaja, Alexandru Ciocarlan, Ion Dragalin, Alic Barba, Nicoleta Vornicu, Elisabeta-Irina Geana, Ionel I. Mangalagiu, Aculina Aricu

**Affiliations:** 1Chemistry of Natural and Biologically Active Compounds Laboratory, Institute of Chemistry, 3 Academiei Str., MD-2028 Chisinau, Moldova; 2Metropolitan Center of Research T.A.B.O.R., 9 Closca Str., RO-700066 Iasi, Romania; 3National Research and Development Institute for Cryogenics and Isotopic Technologies—ICSI Rm. Valcea, 4th Uzinei Str., P.O. Box 7, 240050 Ramnicu Valcea, Romania; 4Faculty of Chemistry, ‘‘Alexandru Ioan Cuza’’ University of Iasi, 11 Carol Bd., RO-700506 Iasi, Romania

**Keywords:** homodrimane sesquiterpenoids, 1,3-benzothiazole, antifungal and antibacterial activities

## Abstract

Based on some homodrimane carboxylic acids and their acyl chlorides, a series of fourteen 2-homodrimenyl-1,3-benzothiazoles, *N*-homodrimenoyl-2-amino-1,3-benzothiazoles, 4′-methyl-homodrimenoyl anilides and 4′-methyl-homodrimenthioyl anilides were synthesized and their biological activities were evaluated on five species of fungi (*Aspergillus niger*, *Fusarium solani*, *Penicillium chrysogenum*, *P. frequentans*, and *Alternaria alternata*) and two strains of bacteria (*Bacillus* sp. and *Pseudomonas aeruginosa*). The synthesis involved the decarboxylative cyclization, condensation and thionation of the said acids, anhydrides or their derivatives with 2-aminothiophenol, 2-aminobenzothiazole, *p*-toluidine and Lawesson’s reagent. As a result, together with the desired compounds, some unexpected products **8**, **25**, and **27** were obtained, and the structures and mechanisms for their formation have been proposed. Compounds **4**, **9**, and **25** showed higher antifungal and antibacterial activity compared to the standards caspofungin (MIC = 1.5 μg/mL) and kanamycin (MIC = 3.0 μg/mL), while compound **8** had comparable activities. In addition, compounds **6**, **17**, and **27** showed selective antifungal activity at MIC = 2.0, 0.25, and 1.0 μg/mL, respectively.

## 1. Introduction

Natural labdane-type diterpenes isolated from terrestrial plants and marine sources are still interesting objects of study due to a wide range of their biological activities [1]. Some of them are obtained from sources in sufficient amounts to be used as precursors for the synthesis of natural analogs, special purpose compounds, or pharmaceutical agents with remarkable properties [2].

The chemistry of 1,3-benzothiazole and its 2-substituted derivatives has become a separate area of research due to a high degree of structural diversity, which generates a wide variety of their applications or pharmacological activities [3,4,5].

A wide variety of reagents and methods which lead to 2-substituted 1,3-benzothiazoles are known. One of the most requested methods of their synthesis involves the cyclocondensation of aromatic aldehydes or others carbonyl compounds such as carboxylic acids, esters, acyl halides, etc., with *o*-aminophenol [6,7] or its disulfides [8]. Frequently, for the conversion of the resulting amides into the corresponding thioamides, Lawesson’s reagent is used, but the course of reactions and their yields strongly depend on the structure of substrates [9].

The synthesis of terpeno-heterocyclic hybrid compounds with a cumulative biological potential is a new direction of organic chemistry that has emerged in the last decade. Research in this field has been successful, a large number of molecular hybrids containing both terpene and diazine [10,11], 1,2,4-triazole and carbazole [12,13], azaheterocyclic [14,15], hydrazinecarbothioamide and 1,2,4-triazole [16], 1,3,4-oxadiazole and 1,3,4-thiadiazole [17], thiosemicarbazone and 1,3-thiazole [18] units were reported, many of which showed excellent antifungal and/or antibacterial activity.

In continuation of our work aimed at the preparation of hybrid terpeno-heterocyclic compounds, herein, we report the result of the synthesis of novel homodrimane sesquiterpenoids bearing 2-substituted 1,3-benzothiazole, *N*-substituted 2-amino-1,3-benzothiazole and *N*-substituted *p*-toluidine units and their antimicrobial properties evaluation.

## 2. Results and Discussion

### 2.1. Synthesis and Characterization

According to the synthesis strategy of the desired compounds, at first, the intermediate carboxylic acids were obtained from commercial (+)-sclareolide (**1**). It was converted to methoxyester **2** in two steps, with an overall yield of 25%, applying the known procedure [19], followed by the saponification into acid **3** in 89% yield. Starting from sclareolide (**1**) carboxylic acids **5** and **7** were obtained in five and six steps, with overall yields of 81% and 62%, respectively [13,20] (Figure 1).

Further, the one-pot decarboxylative cyclization reactions of acids **3**, **5**, and **7** with 2-aminothiophenol promoted by triphenylphosphine and triethylamine [21] were performed under reflux for 4 h, which after silica gel chromatography afforded 2-homodrimenyl-1,3-benzothiazoles **4**, **6**, **8**, and **9**, in the yields as depicted in Figure 1.

The structures of intermediary compounds as well as final products were confirmed by ^1^H, ^13^C, ^15^N, and 2D NMR spectroscopy and HRMS analysis. The formation of the desired hybrid compounds **4**, **6**, **8**, and **9** was proved, first of all, with the presence of signals attributed to aromatic protons from a common 2-substituted-1,3-benzothiazole unit in a range of 7.32–7.96 ppm. In addition, some individual signals such as a singlet corresponding to protons of C_7_-bonded methoxy group at 3.39 ppm, or doublet of doublets of C_6_- and C_7_-bonded protons at 5.88 and 5.95 ppm confirmed the presence of a terpene unit. Those structures were fully confirmed by the carbon spectral data.

It should be noted that, in the case of acid **7**, surprisingly, in addition to the desired compound **9**, obtained with a yield of only 5%, the compound **8** with an unexpected structure was afforded as a major reaction product, in 27% total yield. The rearrangement of the carbon skeleton of compound **8** was confirmed by a shift of some signals in the ^1^H NMR spectrum compared to the starting acid **7**, e.g., by singlet signals of the C_8_- and C_9_-bonded methyl groups at 0.92 and 1.05 ppm and the appearance of new multiplet signals of the C_8_-bonded proton at 1.54–1.56 ppm. The ^13^C NMR spectra confirmed this by signals of the C_8_ (34.5 ppm) and C_9_ (42.1 ppm), those at 130.8 and 139.4 ppm being attributed to C_10_ and C_5_, respectively.

The NMR data of compound **8** have been assigned on the basis of their 1D (^1^H, ^13^C, DEPT-135°) and 2D homo- (^1^H/^13^C HSQC, ^1^H/^13^C HMBC and ^1^H/^1^H COSY-45°) correlation spectra. An analysis of the ^1^H, ^13^C, ^1^H/^1^H COSY and ^1^H/^13^C HSQC NMR spectra suggested the presence of two isolated spin systems: CH_2_CH_2_CH_2_ (C_1_ to C_3_) and CH_2_CH_2_CH (C_6_ to C_8_) (Figure 1). The rearranged carbon framework of compound **8** was indisputable according to a detailed analysis of its ^1^H/^13^C HMBC spectrum. Thus, the observed correlations of H2-C_2_ with two sp^2^ hybridized carbons (C_5_, *δ*_C_ 139.4 and C_10_, *δ*_C_ 130.8) were indicative of the Δ^5,10^ double bond localization, which was also supported by the correlations of H3-C_18_/C_5_, H3-C_9_/C_10_ and H2-C_11_/C_10_. The migration of H3-C_20_ methyl from the C_10_ to C_9_ position was ascertained by the H3-C_20_/C_10_, H3-C_20_/C_8_ and H3-C_20_/C_9_ and the H3-C_20_/C_11_ cross-peaks in the HMBC spectrum.

The rearrangement of 2-homodrimenyl-1,3-benzothiazole **8** carbon skeleton can be explained by the following reaction pathway (Figure 2). The reaction started with formation of triphenylphosphonium chloride as product of Ph_3_P and CCl_4_ interaction, followed by its condensation with carboxylic acid which led to acylphosphonium intermediate **10**. Next, the nucleophilic attack of the amino group to the carbonyl gave the intermediate amide **11** and triphenylphosphine oxide. Then, the nucleophilic attack of the deprotonated sulfur atom led to the unstable cyclic intermediate **12**. Next, the formation of compound **8** was a result of the elimination reaction, which led to the desired 2-homodrimenyl-1,3-benzothiazole **9** that by protonation gave carbocation **13**. The latter suffered a rearrangement of the carbon skeleton as a result of the C_10_-bonded methyl group migration to C_9_, followed by both C_5_ deprotonation and C_5_-C_10_ double bond formation.

Next, a series of new *N*-homodrimenoyl-2-amino-1,3-benzothiazoles were prepared, starting from the intermediate carboxylic acids **3**, **5**, **7**, and **20**, via their acyl chlorides **14**, **16**, **18,** and **21**, generated in situ in 25–65% yields. It should be mentioned that the acid **20** was obtained from the commercially available (+)-sclareolide (**1**) in 6 steps, with an overall yield of 60%, according to the known method [11]. The desired *N*-substituted 2-amino-1,3-benzothiazoles **15**, **17**, **19,** and **22** were obtained with yields between 40–84% by acylation of 2-amino-1,3-benzothiazole with the mentioned sesquiterpene acyl chlorides under the mentioned conditions (Figure 3).

According to the NMR spectra, the hybrids involved both heterocyclic and terpene units, and their accurate masses were confirmed by a high-resolution mass spectrometry (HRMS) analysis. All proton spectra of compounds **15**, **17**, **19**, and **22**, include the signals of aromatic protons in a range of 7.30–7.82 ppm, together with the signals specific for terpene unit such as singlets of C_8_-bonded methyl groups at 1.69–1.82 ppm and C_7_-bonded methoxy group at 3.40 ppm, C_6_- and C_7_-bonded protons at 3.50, 5.93, and 5.94 ppm, singlets of C_17_-exomethylene group at 4.39 and 4.73 ppm, and broad singlets of amidic protons in a range of 9.73–11.26 ppm. The structures of the reported *N*-substituted 2-amino-1,3-benzothiazoles were additionally confirmed by the ^13^C NMR spectra.

Then, effort was devoted to prepare 2-substituted 6-methyl-benzothiazoles starting from the carboxylic acids **3** and **5**, as well as, from acyl chlorides **14**, **16**, **18** using *p*-toluidine. The one-pot condensation of homodrimane acyl chlorides **14**, **16**, and **18**, generated in situ from acids **3**, **5** and **7** (see Figure 3) with *p*-toluidine, yielded amides **23**, **26**, and **28** in 50–54% yields (Figure 4).

An attempt to perform the direct amidation of acids **3** and **5** with *p*-toluidine in the presence of *N*,*N*′-dicyclohexylcarbodiimide (DCC) and 4-dimethylaminopyridine (4-DMAP) gave better results, because the yields of amides **26** and **28** increased up to 76% and 94%, respectively (Figure 4). Together with the signals of protons from terpene units, the proton spectra of amides **23**, **26**, and **28** contained the singlets of C_4′_-bonded methyl in a range of 2.30–2.32 ppm and doublets of aromatic protons from 7.11 ppm to 7.37 ppm, and a broad singlet of the amidic proton at 7.53–7.70 ppm. The structures of the mentioned compounds were fully confirmed by the ^13^C NMR spectra.

After that, amides **23**, **26**, and **28** were submitted to the thionation reaction using Lawesson’s reagent (LR) in toluene [22]. In the case of amide **23**, a reaction occurred, and thioamide **24** was obtained in 50% yield (Figure 4). Its structure was confirmed by the NMR and HRMS analyses. In the ^1^H and ^13^C NMR spectra, down field shifts of the amidic proton to 9.03 ppm and of C_12_ to 200.9 ppm compared to the initial amide **23** were observed.

Heterocyclization of thioamide **24** performed in the presence of potassium ferricyanide under basic conditions (30% NaOH) [22] did not lead to the desired 2-substituted 6-methyl-benzothiazole and gave an unexpected compound **25** in 41% yield (Figure 4). Its structure was elucidated based on the NMR and HRMS spectra. Comparing with the initial thioamide **24**, its ^1^H NMR spectra did not contain the signals of an amidic proton and of one of C_11_-bonded proton, but the singlets of a C_8_-bonded methyl group and one of C_11_-H were shifted to 1.72 and 6.19 ppm, respectively. The same is true of the carbon spectrum, where some signals are strongly shifted, e.g., C_8_ (62.9 ppm), C_11_ (123.6 ppm), C_9_ (170.4 ppm), and C_12_ (176.6 ppm).

The analysis of the structure of compound **25** by ^1^H, ^13^C, ^1^H/^1^H COSY and ^1^H/^13^C HSQC NMR spectra suggested the presence of two isolated spin systems: CH_2_CH_2_CH_2_ (C_1_ to C_3_) and CHCH_2_CH_2_ (C_5_ to C_7_) (Figure 2). In the ^1^H/^13^C HMBC spectrum correlations of H-C_11_ with quaternary carbons (C_9_, *δ*_C_ 170.4 and C_12_, *δ*_C_ 176.6) were observed, which was also supported by the correlations of H-C_11_/C_8_. In addition, the correlation of H3-C_17_/C_12_ confirms the formation of the 5-membered heterocycle.

In the NOESY spectrum of compound **25**, there is no NOE correlation between C_11_-H and C_2′_-H from the aromatic ring (Figure 2) that clearly indicates the *E*-configuration for C_12_ = N- double bond.

The thionation of amide **26** under the same conditions occurred and gave an unexpected cyclic thioamide **27** in 52% yield. The formation of the N-C_8_ bond was confirmed by the absence of an amidic proton signal, a shift of the singlet signal of C_8_-CH_3_ to 1.56 ppm and by the appearance of the C_9_-H doublet at 1.88 ppm. The upfield chemical shift of C_8_ and C_9_ atoms to 59.2 and 58.2 ppm and a downfield shifted signal of C_12_ (175.9 ppm) in the ^13^C NMR spectra also confirmed the structure of compound **27**.

Structural analysis of compound **27** by ^1^H, ^13^C, ^1^H/^1^H COSY and ^1^H/^13^C HSQC NMR spectra suggested the presence of two isolated spin systems: CH_2_CH_2_CH_2_ (C_1_ to C_3_) and CH_2_CHCH (C_5_ to C_7_) (Figure 3). The rearranged carbon framework of compound **27** was indisputable by a detailed analysis of its ^1^H/^13^C HMBC spectrum. Thus, the observed correlations of H-C_5_ with two sp^2^ hybridized carbons (C_6_, *δ*_C_ 126.8 and C_7_, *δ*_C_ 130.6) were indicative of the Δ^6,7^ double bond localization, which was also supported by the correlations of H3-C_17_/C_7_. The relative configuration at C_17_ was deduced from the absence of H3-C_17_/H3-C_10_ NOESY correlation. The position of N has been confirmed by the ^1^H/^15^N HMBC spectra and was supported by the correlations of H2-C_11_/N and H-C_2′_/N cross-peaks (Figure 3).

However, in the case of amide **28**, from the thionation reaction mixture, the same amides **26** and **27** were isolated in 37% and 21% yields, respectively (Figure 4). Their spectral data were in accordance with those obtained earlier.

Returning to the compound **25**, it can be said that its formation in place of the desired benzothiazole is due to the reaction conditions and the molecular structure of thioamide **24** which permits the existence of a tautomeric thioketo-enothiol **24**<—>**29** system. In the basic medium, the tautomer **29** easily generated enothiolate which due to the activation by hexacyanofferate ions attacked the C_8_-C_9_ double bond and generated the intermediate carbocation **30** (Figure 5). In such a way, the formation the new C_8_-S bond occurred simultaneously with the one C_11_-proton elimination giving compound **25**.

The formation of compound **27** can be explained by a sequence of transformations depicted in Figure 6. In this case, the Lawesson’s reagent played a double role. The first role is to interact with axial the C_7_–bonded methoxy group of the amide **28**, stimulating its elimination and generating the carbocation **32** which by deprotonation offered amide **26** (see Figure 4 and Figure 6). On the other hand, the thionation with Lawesson’s reagent gave an intermediate carbocation **33** which suffered a cyclization followed by deprotonation into cyclic thioamide **27**. Note that there are several resonance structures, but from our point of view, intermediates **32** and **33** are more stable.

### 2.2. Antimicrobial Activity

All synthesized compounds were subjected to preliminary screening for their in vitro antifungal and antibacterial activities [23] against pure cultures of fungal species *Aspergillus niger*, *Fusarium solani*, *Penicillium chrysogenum*, *Penicillium frequentans*, and *Alternaria alternata* and both Gram-positive *Bacillus* sp. and Gram-negative *Pseudomonas aeruginosa* bacteria strains. The obtained minimum inhibitory concentration (MIC) values revealed that compounds **4** and **17** possessed a high nonselective antifungal (MIC 0.094 and 0.25 µg/mL, respectively) activity (Table 1, entries 1 and 6) in comparison with caspofungin. Moreover, compounds **6**, **8**, **9**, **25**, and **27** possessed a promising antifungal activity (Table 1, entries 2–4, 11 and 13) at MIC in a range from 0.95 to 2 µg/mL, vs. the same standard. At the same time, compounds **4** and **25** possessed high nonselective antibacterial (MIC 0.75 and 1.5 µg/mL, respectively) activities (Table 1, entries 1 and 11) relative to the standard kanamycin. Compounds **8**, **9**, and **17** possessed a moderate antibacterial activity (Table 1, entries 3, 4, and 11). As to compounds **15**, **19**, **22**, **23**, **24**, **26**, and **28**, they were biologically inactive.

In conclusion, it can be mentioned that the greatest antimicrobial activity was presented by homodrimane sesquiterpenoids bearing benzothiazole units, as well as those containing the NCS fragment, rigidly bound in space with the involvement of another ring, which sterically creates a stable bond similar to that in benzothiazole.

## 3. Materials and Methods

### 3.1. Synthesis and Characterization

The IR spectra were recorded on a Spectrum 100 FT-IR spectrometer (Perkin-Elmer, Shelton, CT, USA) using an ATR technique. The ^1^H, ^13^C, and ^15^N NMR (400, 100, and 40 MHz, respectively) and COSY, ^1^H–^13^C HSQC, ^1^H–^13^C HMBC, DEPT, and ^1^H–^15^N HSQC, ^1^H–^15^N HMBC spectra were acquired on a Bruker Avance DRX 400 spectrometer (Bruker BioSpin, Rheinstetten, Germany) in CDCl_3_ (NMR spectra for all the compounds are available online, see the Appendix A). The ^1^H NMR chemical shifts were reported relative to the residual solvent protons as internal standards (7.26 ppm). The solvent carbon atoms served as internal standard for the ^13^C NMR spectra (77.0 ppm). The ^15^N NMR spectra were obtained using MeNO_2_ (380.5 ppm) and urea (73.4 ppm) as internal standards. Optical rotations measurements were performed on a Jasco DIP-370 polarimeter (Rudolph Research Analytical, Hackettstown, NJ, USA) with a 10 cm microcell. Melting points were determined on a Boetius (VEB Analytik, DDR) hot stage apparatus and were not uncorrected. The progress of reactions and purity of products were examined by TLC on Merck silica gel 60 plates, eluent CH_2_Cl_2_ or a mixture of CH_2_Cl_2_–MeOH, 99:1; 49:1. Visualization was achieved by the treatment with conc. H_2_SO_4_ and heating at 80 °C or using an UV lamp (254 or 365 nm). All solvents were purified and dried by standard techniques prior to use.

Compound **3**: Compound **2** (294 mg, 1 mmol) was dissolved in EtOH (10 mL) and solid KOH (615 mg, 11 mmol) was added. The resulting mixture was heated at 50 °C for 3 h, and then, 2/3 of alcohol was distilled under reduced pressure on a rotary evaporator. The residue was diluted with H_2_O (20 mL), acidified with 40% HCl (20 mL) and extracted with Et_2_O (3 × 20 mL). The organic layer was washed with H_2_O (30 mL), dried over anhydrous Na_2_SO_4_ and concentrated, giving compound **3** (249 mg, 89% yield) as a colorless oil. ${\left[\alpha \right]}_{D}^{20}$ +50.6 (*c* 2.4, CHCl_3_). IR spectrum, ν, cm^−1^: 736, 1070, 1376, 1459, 1626, 1704, 2927. ^1^H NMR (400 MHz, CDCl_3_) *δ* 0.82 (3H, s, 10-C*H*_3_), 0.87 (3H, s, 4-C*H*_3_), 0.88 (3H, s, 4-C*H*_3_), 1.09–1.15 (2H, m, CH_2_), 1.16–1.19 (1H, m, CH_2_), 1.35–1.58 (5H, m, H-5, 2CH_2_), 1.64 (3H, s, 8-C*H*_3_), 1.91–1.94 (1H, m, CH_2_), 3.01 (2H, d, *J* = 7.0 Hz, H-11), 3.32 (3H, s, 7-C*H*_3_), 3.43 (1H, d, *J* = 2.6 Hz, H-7). ^13^C NMR (100 MHz, CDCl_3_) *δ* 17.7 (C-17), 17.9 (C-20), 18.7 (C-2), 21.5 (C-18), 22.5 (C-6), 32.7 (C-19), 32.8 (C-4), 33.0 (C-11), 35.9 (C-1), 39.3 (C-10), 41.1 (C-3), 45.7 (7-OCH_3_), 56.4 (C-5), 79.4 (C-7), 130.0 (C-8), 139.1 (C-9), 171.9 (C-12). HRMS (ESI) calculated for C_17_H_28_O_3_ [M + H]^+^, 280.4035. Found: 280.4127.

Compounds **4**, **6**, **8**, and **9** (General method).

To an ice bath-cooled solution of Ph_3_P (786 mg, 3 mmol) and Et_3_N (0.16 mL, 1.2 mmol) dissolved in CCl_4_ (7 mL), one of the acids **3** (280 mg, 1 mmol), **5** (248 mg, 1 mmol) or **7** (250 mg, 1 mmol) was added. After 10 min of stirring, the solution of 2-aminothiophenol (150 mg, 1.2 mmol) dissolved in CCl_4_ (3 mL) was added and the reaction mixture was refluxed under stirring for 4 h. The solvents were removed under a reduced pressure on a rotary evaporator to dryness and crude reaction products were subjected to silica gel flash chromatography (CH_2_Cl_2_).

Compound **4**. (180 mg, 49%), colorless oil. ${\left[\alpha \right]}_{D}^{20}$ 78.3 (*c* 0.6, CHCl_3_). IR spectrum, ν, cm^−1^: 729, 758, 1080, 1373, 1437, 1456, 1509, 1707, 1759, 2926. ^1^H NMR (400 MHz, CDCl_3_) *δ* 0.86 (3H, s, 10-C*H*_3_), 0.93 (3H, s, 4-C*H*_3_), 0.99 (3H, s, 4-C*H*_3_), 1.09–1.16 (2H, m, CH_2_), 1.23–1.27 (1H, m, CH_2_), 1.35–1.40 (2H, m, CH_2_), 1.51–1.55 (3H, m, H-5, CH_2_), 1.83 (3H, s, 8-C*H*_3_), 2.01–2.05 (1H, m, CH_2_), 3.43 (3H, s, 7-C*H*_3_), 3.51 (1H, t, *J* = 2.7 Hz, 7-CH), 3.88 (2H, t, *J* = 17.4 Hz, H-11), 7.31 (1H, dt, *J* = 7.5, 1.0 Hz, H-6′), 7.43 (1H, dt, *J* = 7.7, 1.0 Hz, H-7′), 7.78 (1H, d, *J* = 7.8 Hz, H-5′), 7.93 (1H, d, *J* = 8.0 Hz, C-8′). ^13^C NMR (100 MHz, CDCl_3_) *δ* 18.6 (C-20 and C-17), 18.7 (C-2), 21.7 (C-18), 22.7 (C-6), 32.8 (C-19), 32.8 (C-11), 32.9 (C-4), 36.0 (C-1), 39.9 (C-10), 41.2 (C-3), 46.0 (7-OCH_3_), 56.9 (C-5), 79.3 (C-7), 121.5 (C-5′), 122.3 (C-8′), 124.4 (C-6′), 125.7 (C-7′), 131.1 (C-8), 135.1 (C-9), 142.5 (C-4′), 153.4 (C-9′), 173.6 (C-2′). ^15^N NMR (400 MHz, CDCl_3_) *δ* 301. HRMS (ESI) calculated for C_23_H_31_NOS [M + H]^+^, 369.5667. Found: 369.5693.

Compound **6**. (185 mg, 55%), yellow oil. ${\left[\alpha \right]}_{D}^{20}$ −260.94 (*c* 0.59, CHCl_3_). IR spectrum, ν, cm^−1^: 729, 757, 1369, 1456, 1508, 1726, 2924. ^1^H NMR (400 MHz, CDCl_3_) *δ* 0.88 (3H, s, 10-C*H*_3_), 0.95 (3H, s, 4-C*H*_3_), 0.96 (3H, s, 4-C*H*_3_), 1.10–1.82 (6H, m, 3CH_2_), 1.86 (3H, s, 8-C*H*_3_), 2.15 (1H, t, *J* = 2.9 Hz, H-5), 3.85 (1H, d, *J* = 16.7 Hz, H-11), 3.96 (1H, d, *J* = 16.7 Hz, H-11), 5.88 (1H, dd, *J* = 9.5 Hz, *J* = 2.5 Hz, H-6), 5.95 (1H, dd, *J* = 9.5, 3.0 Hz, H-7); 7.33 (1H, ddd, *J* = 7.5, 7.2 Hz, *J* = 1.1 Hz, H-6′), 7.44 (1H, ddd, *J* = 8.1, 7.2, 1.2 Hz, H-7′), 7.82 (1H, dm, *J* = 8.3 Hz, H-5′), 7.96 (1H, dm, *J* = 8.1 Hz, H-8′). ^13^C NMR (100 MHz, CDCl_3_) *δ* 15.5 (C-20), 18.5 (C-8), 18.8 (C-2), 22.7 (C-18), 32.1 (C-11), 32.3 (C-19), 32.9 (C-4), 35.1 (C-1), 39.2 (C-10), 40.8 (C-3), 52.9 (C-5), 121.4 (C-5′), 122.4 (C-8′), 124.5 (C-6′), 125.7 (C-7′), 128.7 (C-6), 129.1 (C-7), 135.4 (C-8), 139.9 (C-9), 139.9 (C-9′), 153.6 (C-4′), 174.0 (C-2′). ^15^N NMR (400 MHz, CDCl_3_) *δ* 303. HRMS (ESI) calculated for C_22_H_27_NS [M + H]^+^, 337.1864. Found: 337.1949.

Compound **8**. (92 mg, 27%), mp 58–59 °C, ${\left[\alpha \right]}_{D}^{20}$ 7.63 (*c* 3.2, CHCl_3_). IR spectrum, ν, cm^−1^: 737, 763, 1122, 1311, 1377, 1433, 1454, 1505, 1713, 2920, 3059. ^1^H NMR (400 MHz, CDCl_3_) *δ* 0.92 (3H, d, *J* = 6.8 Hz, 8-C*H*_3_), 1.01 (3H, s, 4-C*H*_3_), 1.02 (3H, s, 4-C*H*_3_), 1.05 (3H, s, 9-C*H*_3_), 1.30–1.40 (2H, m, CH_2_), 1.46–1.63 (5H, m, H-8, 2CH_2_), 1.87–2.04 (2H, m, H-6), 2.17–2.22 (2H, m, CH_2_), 3.27 (2H, d, *J* = 3.9 Hz, H-11), 7.32 (1H, td, *J* = 7.8, 1.2 Hz, H-6′), 7.42 (1H, td, *J* = 7.0, 1.2 Hz, H-7′), 7.83 (1H, dm, *J* = 8.0 Hz, H-5′), 7.96 (1H, d, *J* = 8.0 Hz, H-8′). ^13^C NMR (100 MHz, CDCl_3_) *δ* 16.2 (C-17), 20.1 (C-2), 21.4 (C-20), 24.9 (C-6), 26.6 (C-1), 27.0 (C-7), 27.8 (C-18), 28.3 (C-19), 34.5 (C-8); 34.7 (C-4), 39.6 (C-3), 42.1 (C-11, C-9), 121.3 (C-5′), 122.5 (C-8′), 124.5 (C-6′), 125.6 (C-7′), 130.8 (C-10), 135.8 (C-9′), 139.4 (C-5), 152.1 (C-4′), 169.5 (C-2′). ^15^N NMR (400 MHz, CDCl_3_) *δ* 305. HRMS (ESI) calculated for C_22_H_29_NS [M + H]^+^, 339.2021. Found: 339.2106.

Compound **9**. (17 mg, 5%), yellow oil. ${\left[\alpha \right]}_{D}^{20}$ +28.62 (*c* 0.2, CHCl_3_). IR spectrum, ν, cm^−1^: 729, 757, 1014, 1124, 1146, 1293, 1376, 1435, 1456, 1506, 1577, 1694, 2865, 2926. ^1^H NMR (400 MHz, CDCl_3_) *δ* 0.84 (3H, s, 4-C*H*_3_), 0.90 (3H, s, 4-C*H*_3_), 1.02 (3H, s, 10-C*H*_3_), 1.05–1.11 (2H, m, CH_2_), 1.25–1.40 (3H, m, H-5, CH_2_), 1.45–1.60 (4H, m, 2CH_2_), 1.70 (3H, s, 8-C*H*_3_), 2.09–2.21 (2H, m, H-7), 3.83 (1H, d, *J* = 17.2 Hz, H-11), 3.90 (1H, d, *J* = 17.2 Hz, H-11), 7.32 (1H, ddd, *J* = 7.9 Hz, *J* = 7.3, 1.1 Hz, H-6′), 7.43 (1H, ddd, *J* = 8.2, 7.3, 1.3 Hz, H-7′), 7.81 (1H, ddd, *J* = 7.9, 1.3, 0.5 Hz, H-5′), 7.94 (1H, d, *J* = 8.2 Hz, H-8′). ^13^C NMR (100 MHz, CDCl_3_) *δ* 18.8 (C-6), 18.9 (C-2), 20.3 (C-20), 20.7 (C-17), 21.6 (C-18), 32.8 (C-11), 33.2 (C-19), 33.3 (C-4); 33.6 (C-7), 36.6 (C-1), 39.1 (C-10), 41.6 (C-3), 51.8 (C-5), 121.3 (C-5′), 122.4 (C-8′), 124.3 (C-6′), 125.6 (C-7′), 131.2 (C-8), 135.4 (C-9′), 137.7 (C-9), 153.6 (C-4′), 175.4 (C-2′). ^15^N NMR (400 MHz, CDCl_3_) *δ* 299. HRMS (ESI) calculated for C_22_H_29_NS [M + H]^+^, 339.2021. Found: 339.2107.

Compounds **15**, **17**, **19**, and **22** (General method).

The solution of one of the acids **3** (280 mg, 1 mmol), **5** (248 mg, 1 mmol), **7** (250 mg, 1 mmol) or **20** (250 mg, 1 mmol) dissolved in anhydrous C_6_H_6_ (5 mL) was treated with a solution of (COCl)_2_ (0.95 mL, 11 mmol) dissolved in C_6_H_6_ (2.5 mL). The reaction mixture was stirred at room temperature for 1 h and subsequently refluxed for 1 h. The C_6_H_6_ and excess of (COCl)_2_ were removed at a reduced pressure on a rotary evaporator. Next, 2-aminobenzothiazole (225 mg, 1.5 mmol) was added to the solution of an acyl chloride **14**, **16**, **18** or **21** in CH_2_Cl_2_ (10 mL), and the resulting mixtures were stirred at r.t. for 3 h, then refluxed for 4–10 h. After cooling, the precipitates were filtered off, washed with CH_2_Cl_2_, and the filtrates were concentrated to dryness at a reduced pressure on a rotary evaporator. The crude reaction products were purified by silica gel flash chromatography (1 → 2% MeOH-CH_2_Cl_2_).

Compound **15**. (164 mg, 40%), white crystals, mp 93–94 °C. ${\left[\alpha \right]}_{D}^{20}$ 99.96 (*c* 1.4, CHCl_3_). IR spectrum, ν, cm^−1^: 728, 755, 1076, 1341, 1264, 1441, 1538, 1600, 1697, 2926, 3182. ^1^H NMR (400 MHz, CDCl_3_) *δ* 0.85 (3H, s, 10-C*H*_3_), 0.91 (3H, s, 4-C*H*_3_), 0.93 (3H, s, 4-C*H*_3_); 1.09–1.20 (2H, m, CH_2_), 1.37–1.58 (5H, m, H-5, 2CH_2_), 1.62–1.66 (1H, m, CH_2_), 1.77 (3H, s, 8-C*H*_3_), 2.01 (1H, d, *J* = 13.0 Hz, CH_2_), 3.22 (1H, d, *J* = 17.4 Hz, H-11), 3.32 (1H, d, *J* = 17.4 Hz, H-11), 3.40 (3H, s, 7-C*H*_3_), 3.50 (1H, d, *J* = 5.6 Hz, H-7), 7.30 (1H, dt, *J* = 7.5, 0.9 Hz, H-6′), 7.42 (1H, dt, *J* = 7.6, 1.0 Hz, H-7′), 7.77 (1H, d, *J* = 8.0 Hz, H-5′), 7.80 (1H, d, *J* = 7.7 Hz, H-8′), 9.91 (1H, br.s, NH). ^13^C NMR (100 MHz, CDCl_3_) *δ* 18.2 (C-20), 18.3 (C-17), 18.7 (C-2), 21.6 (C-18), 22.4 (C-6), 32.7 (C-19), 32.8 (C-4), 35.7 (C-11), 35.8 (C-1), 39.6 (C-10), 40.9 (C-3), 45.5 (7-C*H*_3_), 56.8 (C-5), 78.9 (C-7), 120.8 (C-5′), 121.3 (C-8′), 123.9 (C-6′), 126.2 (C-7′), 132.1 (C-4′), 132.7 (C-8), 138.9 (C-9), 148.1 (C-9′), 158.2 (C-2′), 169.6 (C = O). ^15^N NMR (400 MHz, CDCl_3_) *δ* 140, 259. HRMS (ESI) calculated for C_24_H_32_N_2_O_2_S [M-H]^-^, 412.2184. Found: 412.2112.

Compound **17**. (23 mg, 52%), white crystals, mp 83–84 °C. ${\left[\alpha \right]}_{D}^{20}$ −172.5 (*c* 0.1, CHCl_3_). IR spectrum, ν, cm^−1^: 728, 755, 908, 1147, 1267, 1343, 1442, 1536, 1599, 1702, 2925, 3178. ^1^H NMR (400 MHz, CDCl_3_) *δ* 0.84 (3H, s, 10-C*H*_3_), 0.96 (6H, s, 4-C*H*_3_, 4-C*H*_3_), 1.15–1.80 (6H, m, 3CH_2_), 1.83 (3H, s, 8-C*H*_3_), 2.10 (1H, t, *J* = 2.2 Hz, H-5), 3.21 (1H, d, *J* = 17.3 Hz, H-11), 3.42 (1H, d, *J* = 17.0 Hz, H-11), 5.93 (1H, d, *J* = 1.8 Hz, H-6), 5.94 (1H, d, *J* = 2.2 Hz, H-6), 7.31 (1H, dt, *J* = 1.0, 0.8 Hz, H-6′), 7.44 (1H, dt, *J* = 1.2, 1.0 Hz, H-7′), 7.75 (1H, d, *J* = 8.2 Hz, H-5′), 7.82 (1H, dd, *J* = 7.87, 0.47 Hz, H-8′), 9.74 (1H, br.s, NH). ^13^C NMR (100 MHz, CDCl_3_) *δ* 15.1 (C-20), 18. (C-17), 18.7 (C-2), 22.8 (C-18), 32.2 (C-19), 33.0 (C-4), 35.0 (C-1), 35.3 (C-11), 39.1 (C-10), 40.6 (C-3), 52.9 (C-5), 120.7 (C-5′), 121.4 (C-8′), 124.0 (C-6′), 126.3 (C-7′), 128.8 (C-6), 129.6 (C-7), 130.9 (C-9), 132.1 (C-4′), 135.7 (C-8), 148.0 (C-9′), 157.9 (C-2′), 169.9 (C-12). ^15^N NMR (400 MHz, CDCl_3_) *δ* 139, 254. HRMS (ESI) calculated for C_23_H_28_N_2_OS [M + H]^+^, 380.1922. Found: 380.2011.

Compound **19.** (171 mg, 45%), white crystals, mp 84–85 °C. ${\left[\alpha \right]}_{D}^{20}$ +102.5 (*c* 2.0, CHCl_3_). IR spectrum, ν, cm^−1^: 727, 755, 883, 907, 975, 1018, 1152, 1267, 1334, 1379, 1443, 1533, 1600, 1704, 1773, 2927, 3175, 3365. ^1^H NMR (400 MHz, CDCl_3_) *δ* 0.80 (3H, s, 4-C*H*_3_), 0.84 (3H, s, 4-C*H*_3_), 0.88 (3H, s, 10-C*H*_3_), 0.95–1.18 (4H, m, 2CH_2_), 1.27 (1H, dd, *J* = 12.6 Hz, *J* = 1.9 Hz, H-5), 1.33–1.57 (4H, m, 2CH_2_), 1.67 (3H, s, 8-C*H*_3_), 2.09 (1H, dd, *J* = 18.0 Hz, *J* = 6.4 Hz, H-7), 2.27 (1H, ddd, *J* = 18.4 Hz, *J* = 11.2 Hz, *J* = 7.4 Hz, H-7), 3.12 (1H, d, *J* = 17.6 Hz, H-11), 3.33 (1H, d, *J* = 17.6 Hz, H-11), 7.30 (1H, ddd, *J* = 8.0 Hz, *J* = 7.3 Hz, *J* = 1.0 Hz, H-7′), 7.42 (1H, ddd, *J* = 8.0 Hz, *J* = 7.3 Hz, *J* = 1.2 Hz, H-6′), 7.79 (1H, d, *J* = 8.0 Hz, H-5′), 7.81 (1H, ddd, *J* = 8.0 Hz, *J* = 1.2 Hz, *J* = 0.6 Hz, H-8′), 9.73 (1H, br.s., NH). ^13^C NMR (100 MHz, CDCl_3_) *δ* 18.7 (C-6), 18.8 (C-2), 19.7 (C-20), 20.4 (C-17), 21.6 (C-18), 33.1 (C-19), 33.3 (C-4), 33.5 (C-7), 35.9 (C-11), 36.3 (C-1), 38.8 (C-10), 41.3 (C-3), 51.5 (C-5), 120.7 (C-5′), 121.5 (C-8′), 124.0 (C-7′), 126.3 (C-6′), 132.1 (C-9′), 133.0 (C-9), 134.2 (C-8), 148.0 (C-4′), 158.4 (C-2′), 170.4 (C-12). ^15^N NMR (400 MHz, CDCl_3_) *δ* 137, 257. HRMS (ESI) calculated for C_23_H_30_N_2_OS [M + H]^+^, 382.2079. Found: 382.2165.

Compound **22**. (320 mg, 84%), white crystals, mp 99–100 °C. ${\left[\alpha \right]}_{D}^{20}$ −34.2 (*c* 2.0, CHCl_3_). IR spectrum, ν, cm^−1^: 750, 884, 998, 1018, 1135, 1157, 1215, 1270, 1292, 1324, 1332, 1365, 1383, 1443, 1457, 1542, 1599, 1644, 1697, 2932, 3063, 3178. ^1^H NMR (400 MHz, CDCl_3_) *δ* 0.46 (3H, s, 10-C*H*_3_), 0.76 (3H, s, 4-C*H*_3_), 0.84 (3H, s, 4-C*H*_3_), 0.99–1.18 (3H, m, H-5, CH_2_), 1.25 (1H, dd, *J* = 13.0, 4.4 Hz, H-6), 1.30–1.54 (4H, m, 2CH_2_), 1.68 (1H, dm, *J* = 13.0 Hz, H-6), 2.05 (1H, td, *J* = 13.0, 5.1 Hz, H-7), 2.33 (1H, ddd, *J* = 13.0, 4.0, 2.1 Hz, H-7), 2.45 (1H, dd, *J* = 11.2, 10.3 Hz, H-9), 2.47 (1H, dd, *J* = 6.2, 10.3 Hz, H-11), 2.63 (1H, dd, *J* = 26.2, 11.2 Hz, H-11), 4.39 (1H, s, 8-CH_2_), 4.73 (1H, s, 8-CH_2_), 7.33 (1H, ddd, *J* = 8.1, 7.1, 1.0 Hz, H-7′), 7.44 (1H, ddd, *J* = 8.2, 7.1, Hz, 1.1 Hz, H-6′), 7.81 (1H, dd, *J* = 8.2, 1.0 Hz, H-5′), 7.84 (1H, dd, *J* = 8.1, 1.1 Hz, H-8′), 11.26 (1H, br.s., NH). ^13^C NMR (100 MHz, CDCl_3_) *δ* 14.3 (C-20), 19.2 (C-2); 21.6 (C-18), 23.9 (C-6), 32.8 (C-11), 33.4 (C-4), 33.5 (C-19), 37.5 (C-7), 38.9 (C-1), 39.0 (C-10), 41.8 (C-3), 52.2 (C-9), 55.0 (C-5), 106.5 (8-CH_2_), 120.6 (C-5′), 121.7 (C-8′), 123.9 (C-7′), 126.3 (C-6′), 132.1 (C-9′), 147.8 (C-4′), 148.7 (C-8), 159.6 (C-2′), 171.8 (C-12). ^15^N NMR (400 MHz, CDCl_3_) *δ* 251 (C = N). HRMS (ESI) calculated for C_23_H_30_N_2_OS [M + H]^+^, 382.2079. Found: 382.2166.

Compounds **23**, **26**, and **28**. (Typical procedure).

*Method 1.* The solution of one of the acids **3** (280 mg, 1 mmol), **5** (248 mg, 1 mmol) or **7** (250 mg, 1 mmol) dissolved in anhydrous C_6_H_6_ (5 mL) was treated with a solution of (COCl)_2_ (0.95 mL, 11 mmol) dissolved in C_6_H_6_ (2.5 mL). The reaction mixture was stirred at room temperature for 1 h, and additionally refluxed for 1 h. The C_6_H_6_ and excess of (COCl)_2_ were removed at reduced pressure on a rotary evaporator. Next, *p*-toluidine (160 mg, 1.5 mmol) was added to the solutions of acyl chlorides **14**, **16** or **18** obtained in situ in CH_2_Cl_2_ (10 mL), and the resulting mixtures were stirred at r.t. for 5 h, and refluxed for 10–12 h. After cooling, the precipitate was filtered off, washed with CH_2_Cl_2_, and the filtrate was concentrated to dryness on a rotary evaporator. The crude reaction products were purified by silica gel flash chromatography (1→2% MeOH-CH_2_Cl_2_) to give products **23**, **26**, and **28**.

*Method 2.* A solution of DCC (412 mg, 2 mmol), 4-DMAP (244 mg, 2 mmol), *p*-toluidine (214 mg, 2 mmol) and an acid **3** (280 mg, 1 mmol) or **5** (248 mg, 1 mmol) dissolved in CH_2_Cl_2_ (8 mL) was stirred for 10 h at room temperature. After the reaction period, the mixture was filtered, and the solvent was removed under a reduced pressure on a rotary evaporator to give the crude product which was purified by silica gel flash chromatography (CH_2_Cl_2_) to give compounds **26** and **28**.

Compound **23**. (183 mg, 54%), white crystals, mp 152–153 °C. ${\left[\alpha \right]}_{D}^{20}$ +132.64 (*c* 1.0, CHCl_3_). IR spectrum, ν, cm^−1^: 733, 818, 908, 1171, 1248, 1346, 1405, 1458, 1516, 1603, 1661, 2867, 2927, 3293. ^1^H NMR (400 MHz, CDCl_3_) *σσ* 0.85 (3H, s, 4-C*H*_3_), 0.92 (3H, s, 4-C*H*_3_), 0.99 (3H, s, 10-C*H*_3_), 1.04–1.10 (2H, m, CH_2_), 1.17 (1H, dd, *J* = 12.6, 1.9 Hz, H-5), 1.40–1.62 (4H, m, 2CH_2_), 1.68 (3H, s, 8-C*H*_3_), 1.71–1.82 (2H, m, CH_2_), 2.08–2.26 (2H, m, H-7), 2.31 (3H, s, 4′-C*H*_3_), 3.04 (1H, d, *J* = 17.5 Hz, H-11), 3.22 (1H, d, *J* = 17.5 Hz, H-11), 7.12 (2H, d, *J* = 8.2 Hz, H-3′ and H-5′), 7.35 (2H, d, *J* = 8.3 Hz, H-2′ and H-6′), 7.53 (1H, br.s, NH). ^13^C NMR (100 MHz, CDCl_3_) *δ* 18.8 (C-6), 18.9 (C-2), 20.1 (C-20), 20.3 (C-17), 21.7 (C-18), 29.9 (C-7′), 33.3 (C-19), 33.4 (C-4), 33.7 (C-7), 36δ.3 (C-1), 37.2 (C-11), 39.0 (C-10), 41.6 (C-3), 52.4 (C-5), 119.9 (C-2′ and C-6′), 129.5 (C-3′ and C-5′), 132.1 (C-8), 133.9 (C-4′), 135.3 (C-1′), 136.4 (C-9), 169.6 (C-12). ^15^N NMR (400 MHz, CDCl_3_) *δ* 127. HRMS (ESI) calculated for C_23_H_33_NO [M + H]^+^, 339.2562. Found: 339.2649.

Compound **26**. (256 mg, 76%), white crystals, mp 69–70 °C, ${\left[\alpha \right]}_{D}^{20}$ −155.56 (*c* 0.69, CHCl_3_). IR spectrum, ν, cm^−1^: 817, 1177, 1607, 1245, 1351, 1454, 1513, 1542, 1653, 2927, 3292. ^1^H NMR (400 MHz, CDCl_3_) *δ* 0.87 (3H, s, 10-C*H*_3_), 0.97 (3H, s, 4-C*H*_3_), 0.99 (3H, s, 4-C*H*_3_), 1.10–1.62 (6H, m, 3CH_2_), 1.83 (3H, s, 8-C*H*_3_), 2.04 (1H, t, *J* = 2.5 Hz, H-5), 2.32 (3H, s, 4′-C*H*_3_), 3.08 (1H, d, *J* = 16.9 Hz, H-11), 3.31 (1H, d, *J* = 16.9 Hz, H-11),5.92 (1H, dd, *J* = 9.4, 2.4 Hz, H-6), 5.97 (1H, dd, *J* = 9.5, 2.7 Hz, H-7), 7.14 (2H, d, *J* = 8.2 Hz, H-2′ and H-6′), 7.37 (2H, d, *J* = 8.4 Hz, H-3′ and H-5′), 7.66 (1H, s, NH). ^13^C NMR (100 MHz, CDCl_3_) *δ* 15.0 (C-17), 18.4 (C-20), 18.7 (C-2); 20.8 (C-7′); 22.7 (C-18), 32.4 (C-19), 33.0 (C-4), 34.8 (C-1), 36.5 (C-11), 39.1 (C-10), 40.8 (C-3), 53.6 (C-5), 119.9 (C-2′ and C-6′), 128.9 (C-6), 129.4 (C-7), 129.7 (C-3′and C-5′), 129.9 (C-8), 135.1 (C-9), 138.1 (C-1′), 169.0 (C-12). ^15^N NMR (400 MHz, CDCl_3_) *δ* 125. HRMS (ESI) calculated for C_23_H_31_NO [M + H]^+^, 337.2406. Found: 337.2492.

Compound **28**. (346 mg, 94%), white solid, mp 187–188 °C, ${\left[\alpha \right]}_{D}^{20}$ +70.2 (*c* 0.39, CHCl_3_). IR spectrum, ν, cm^−1^: 816, 1086, 1243, 1310, 1448, 1536, 1571, 1624, 2928, 3322. ^1^H NMR (400 MHz, CDCl_3_) *δ* 0.87 (3H, s, 10-C*H*_3_), 0.93 (3H, s, 4-C*H*_3_), 0.95 (3H, s, 4-C*H*_3_), 1.10–1.19 (2H, m, CH_2_), 1.39–1.60 (5H, m, H-5 and 2CH_2_), 1.78 (3H, s, 8-C*H*_3_), 2.00–2.04 (2H, m, CH_2_), 2.30 (3H, s, 4′-C*H*_3_), 3.05 (1H, d, *J* = 17.6 Hz, H-11), 3.22 (1H, d, *J* = 17.6 Hz, H-11), 3.39 (3H, s, 7-C*H*_3_), 3.48 (1H, d, *J* = 2.6 Hz, H-7), 7.11 (2H, d, *J* = 8.2 Hz, H-2′ and H-6′), 7.35 (2H, d, *J* = 8.4 Hz, H-3′ and H-5′), 7.70 (1H, s, NH). ^13^C NMR (100 MHz, CDCl_3_) *δ* 18.2 (C-20), 18.4 (C-2), 18.6 (C-17), 20.8 (C-7′), 21.6 (C-18), 22.4 (C-6), 32.8 (C-19), 32.9 (C-4), 35.4 (C-1), 37.0 (C-11), 39.7 (C-10), 41.2 (C-3), 45.9 (7-OCH_3_), 56.8 (C-5), 79.0 (C-7), 120.1 (C-2′ and C-6′), 129.3 (C-3′ and 5′), 132.0 (C-8), 133.3 (C-4′), 135.1 (C-9), 140.8 (C-1′), 168.6 (C-12). ^15^N NMR (400 MHz, CDCl_3_) *δ* 128. HRMS (ESI) calculated for C_24_H_35_NO_2_ [M-31]^+^, 369.2668. Found: 338.2492.

Compounds **24** and **27**. (Typical procedure)

To a solution of one of the amides **23** (339 mg, 1 mmol), **26** (337 mg, 1 mmol) or **28** (369 mg, 1 mmol) dissolved in toluene (8 mL), Lawesson’s reagent (203 mg, 0.5 mmol) was added and the reaction mixture was refluxed for 48–50 h. Then, the mixture was filtered, and the solvent was removed under a reduced pressure on a rotary evaporator to afford the crude reaction product, which was purified by silica gel flash column chromatography (1% MeOH-CH_2_Cl_2_).

Compound **24**. (177 mg, 50%), white solid, mp 104–105 °C. ${\left[\alpha \right]}_{D}^{20}$ +45.63 (*c* 0.5, CHCl_3_). IR spectrum, ν, cm^−1^: 730, 826, 852, 908, 998, 1056, 1066, 1267, 1395, 1406, 1453, 1516, 1599, 2052, 2214, 2972, 2987, 3147, 3246. ^1^H NMR (400 MHz, CDCl_3_) *δ* 0.86 (3H, s, 4-C*H*_3_), 0.91 (3H, s, 4-C*H*_3_), 1.00 (3H, s, 10-C*H*_3_), 1.03–1.14 (2H, m, CH_2_), 1.15 (1H, dd, *J* = 12.6, 2.0 Hz, H-5), 1.34–1.59 (4H, m, 2CH_2_), 1.67 (3H, s, 8-C*H*_3_), 1.72–1.88 (2H, m, CH_2_), 2.11–2.23 (2H, m, H-7), 2.36 (3H, s, 4′-C*H*_3_), 3.71 (2H, s, H-11), 7.22 (2H, d, *J* = 8.4 Hz, H-3′ and H-5′), 7.50 (2H, d, *J* = 8.4 Hz, H-2′ and H-6′), 9.03 (1H, s, NH). ^13^C NMR (100 MHz, CDCl_3_) *δ* 18.7 (C-2), 18.8 (C-6), 20.1 (C-20), 20.2 (C-17), 21.1 (4′-C*H*_3_), 21.6 (C-18), 33.2 (C-19), 33.4 (C-4), 33.6 (C-7), 36.2 (C-1), 39.2 (C-10), 41.6 (C-3), 47.8 (C-11), 52.5 (C-5), 123.8 (C-2′ and C-6′), 129.5 (C-3′ and C-5′), 134.2 (C-8), 136.2 (C-4′), 136.8 (C-9), 136.9 (C-1′), 200.9 (C = S). HRMS (ESI) calculated for C_23_H_33_NS [M + H]^+^, 355.2334. Found: 355.2419.

Compound **27**. (176 mg, 52%), white solid, mp 103–105 °C. ${\left[\alpha \right]}_{D}^{20}$ +155.73 (*c* 0.83, CHCl_3_). IR spectrum, ν, cm^−1^: 817, 1033, 1097, 1365, 1454, 1502, 1600, 1625, 2917. ^1^H NMR (400 MHz, CDCl_3_) *δ* 0.82 (3H, s, 4-C*H*_3_), 0.87 (3H, s, 4-C*H*_3_), 0.92 (3H, s, 10-C*H*_3_), 1.08–1.40 (2H, m, CH_2_), 1.46–1.55 (2H, m, CH_2_), 1.56 (3H, s, 8-C*H*_3_), 1.64–1.72 (2H, m, CH_2_), 1.88 (1H, d, *J* = 8.2 Hz, H-9), 2.23 (3H, s, H-7′), 3.07 (1H, d, *J* = 17.5 Hz, H-11), 3.19 (1H, dd, *J* = 17.5, 8.2 Hz, H-11), 5.62 (1H, dd, *J* = 10.1, 2.1 Hz, H-6), 5.66 (1H, dd, *J* = 10.1, 1.2 Hz, H-7), 6.75 (2H, d, *J* = 8.0 Hz, H-2′ and H-6′), 7.11 (2H, d, *J* = 8.0 Hz, H-3′ and H-5′). ^13^C NMR (100 MHz, CDCl_3_) *δ* 14.5 (C-20), 18.2 (C-2), 20.9 (C-7′), 21.6 (C-18), 32.5 (C-4), 33.9 (C-19), 33.9 (C-17), 37.6 (C-1), 37.7 (C-10), 40.7 (C-11), 40.9 (C-3), 51.9 (C-5), 58.2 (C-9), 59.2 (C-8), 119.9 (C-2′ and C-6′), 126.8 (C-6), 129.6 (C-3′ and C-5′), 130.6 (C-7), 133.4 (C-4′), 149.8 (C-1′), 175.9 (C-12). ^15^N NMR (400 MHz, CDCl_3_) *δ* 293. HRMS (ESI) calculated for C_23_H_31_NS [M + H]^+^, 353.5677. Found: 353.5748.

Compound **25**. To a solution of carbothioamide **24** (355 mg, 1 mmol) dissolved in EtOH (9 mL), 30% NaOH (1 mL, 7.9 mmol) was added. The mixture was diluted with EtOH (20 mL) to give 10% NaOH. Portions of this mixture were added to a stirred solution of K_3_[Fe(CN)_6_] (1.3 g, 3.9 mmol) in H_2_O (2 mL) at 85 °C. The resultant mixture was further heated at 85 °C for 5 h and filtered to isolate a light yellow solid (720 mg) K_4_[Fe(CN)_6_]·3H_2_O. Then, the solvent was removed in vacuo from the filtrate. To the residue, H_2_O (20 mL) was added and the obtained mixture was extracted with CH_2_Cl_2_ (2 × 30 mL). The combined extracts were washed with H_2_O (2 × 20 mL), dried over MgSO_4_, and the solvent was removed to afford an orange oil. The crude reaction product was purified by flash column chromatography (SiO_2_, elution CH_2_Cl_2_) to give compound **25**.

Compound **25**. (145 mg, 41%), yellow oil. ${\left[\alpha \right]}_{D}^{20}$ −92.68 (*c* 2.0, CHCl_3_). IR spectrum, ν, cm^−1^: 730, 824, 851, 909, 1004, 1127, 1147, 1169, 1201, 1249, 1306, 1379, 1451, 1504, 1594, 1618, 2868, 2927. ^1^H NMR (400 MHz, CDCl_3_) *δ* 0.89 (3H, s, 4-C*H*_3_), 0.91 (3H, s, 4-C*H*_3_), 1.07 (1H, dd, *J* = 12.5 Hz, *J* = 2.5 Hz, H-5), 1.19–1.22 (1H, m, CH_2_), 1.23 (3H, s, 10-CH_3_), 1.42–1.50 (2H, m, CH_2_), 1.58–1.65 (2H, m, CH_2_), 1.72 (3H, s, 8-C*H*_3_), 1.78–1.95 (4H, m, 2CH_2_), 2.22–2.27 (1H, m, CH_2_), 2.33 (3H, s, 4′-C*H*_3_), 6.19 (1H, s, H-11), 6.98 (2H, d, *J* = 8.0 Hz, H-3′ and H-5′), 7.14 (2H, d, *J* = 8.0 Hz, H-2′ and H-6′). ^13^C NMR (100 MHz, CDCl_3_) *δ* 18.6 (C-6); 19.6 (C-20), 19.9 (C-2), 21.0 (C-7′), 21.6 (C-18), 29.4 (C-17), 33.3 (C-19), 34.0 (C-4), 39.0 (C-1), 41.2 (C-10), 41.7 (C-3), 43.2 (C-7), 55.2 (C-5), 62.9 (C-8), 120.5 (C-2′ and C-6′), 123.6 (C-11), 129.6 (C-3′ and C-5′), 134.0 (C-4′), 148.9 (C-1′), 170.4 (C-9), 176.6 (C-12). ^15^N NMR (400 MHz, CDCl_3_) *δ* 191. HRMS (ESI) calculated for C_23_H_31_NS [M + H]^+^, 353.5677. Found: 353.5725.

### 3.2. Antifungal and Antibacterial Activity Assay

Pure cultures of the fungi *Aspergillus niger*, *Fusarium*, *Penicillium chrysogenum*, *Penicillium frequentans*, and *Alternaria alternata* and bacteria *Pseudomonas aeruginosa* and *Bacillus* sp. were obtained from the American Type Culture Collection (ATCC). Suspensions of microorganisms in DMSO were prepared according to direct colony method and serial dilution procedure. Then, the final concentration of the stock inoculum was 1·10^−4^ μg/mL. Both antifungal and antibacterial activity assay were performed by applying a mixture of a microorganism suspension and a solution of the target compound in a ratio 1:1 to Petri dishes with a solid medium—Merck Sabouraud agar or agar-agar. The DMSO did not have any inhibitory effect on the tested organisms.

## 4. Conclusions

A series of 14 novel hybrid terpeno-heterocyclic compounds containing homodrimane and 2-substituted 1,3-thiadiazole, *N*-substituted 2-amino-1,3-benzothiazole and *N*-*p*-toluidyl units were designed, synthesized, and assessed as antimicrobial agents. Several of them showed higher antifungal and antibacterial activities than reference drugs.

## Data Availability

Not applicable.

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
