# Peer review of "Synthesis of Homodrimane Sesquiterpenoids Bearing 1,3-Benzothiazole Unit and Their Antimicrobial Activity Evaluation"

_molecules, 2022, doi:10.3390/molecules27165082_

Round 1

Reviewer 1 Report

Although the manuscript by Aricu et al. involves some interesting results with full experimental data, it should not be accepted without major revision due to the following reason.

1.     Reaction pathways in Schemes 2 and 5 are not reasonable.

2.     This manuscript involves many wrong expressions in English and should be re-written with the aid of English editing service.

Please describe reaction temperature in Schemes 1, 3 and 4 instead of delta.

Please mention on the role of PPh3 in Scheme 2.

In Scheme 2, authors suggested initial formation of the amide. However, nucleophilicity of the nitrogen in aniline is not high enough to provide amide under the reaction conditions. Alternative pathway might be initial formation of thioester followed by the attack of the nitrogen lone pair to the carbonyl group. Please re-consider on the reaction pathway of benzothioimidazole synthesis.

In Scheme 5, departure of a hydride from the intermediate 28 is not reasonable under the reaction conditions. Abstraction of the hydrogen as a proton by a conjugate might be feasible.

In the intermediate 29, the upper arrow indicates intramolecular hydride attack to the olefin, which is not acceptable. The lower arrow indicates the attack of the nitrogen lone pair to the double bond, which is not reasonable. Activation of the double bond might be needed to allow such nucleophilic attack. Please re-consider.   

Determination of the geometry of the C=N bond of compound 24 is needed. 

The English in this manuscript should be revised and edited by a native speaker. 

Lack of definite article “the” gives difficulty in catching the meaning of sentences.

Following corrections are suggested, though not sufficient corrections.

Authors should be careful to tense, plural etc.   

27: compared –> compared to

63: 1 –> (1)

64: the same initial lactone –> sclrareoride (1), 

64: delete “others intermediate”   

65: delete “separate”

68: was –> were

71: depicted –> in the yields as depicted in the Scheme 1.

77: was –> were

77: signal –> signals

82: spectra pattern –> spectral data

83: instead –> in addition to 

85: obtained –> afforded

86: spectra –> spectrum

87: initial –> starting

91: base –> basia

95: at –> according to

105: mechanism –> Reaction pathway

114: ditto

118: 1 –> (1)

123: contains –> involve

162: signal –> signals

164: of compound 23 –> –> of the structure of compound 24 by

173: in –> under

177: confirm –> confirmed

179: analysis –> structural analysis

179: of –> by

202: mechanism –> reaction pathway

204: first of all –> initial

205: delete “and led to obtained before amide”

213: reveal –> revealed

214: posses –> possessed

216: ditto

217: ditto

220: ditto

221: are –> were

244: remain –> residue

244: acidulated –> acidified 

313: additionally –> subsequently

315: delete “obtained in situ”

316: delete “dissolved”

Author Response

Comments/suggestions:

Answers/corrections:

Reaction pathways in Schemes 2 and 5 are not reasonable.

The reaction pathways were revised. The Scheme 5 was divided in Schemes 5 and 6, respectively. This is our vision on reaction mechanisms, based on our experience in the field. But we did not exclude other explications

This manuscript involves many wrong expressions in English and should be re-written with the aid of English editing service.

The manuscript was re-writen and edited by a specialist in the field.

Please describe reaction temperature in Schemes 1, 3 and 4 instead of delta.

The reaction temperatures were mentioned in Schemes 1, 3 and 4.

Please mention on the role of PPh3 in Scheme 2.

PPh3 promotes the condensation reaction between 2-aminothiophenol and carboxylic acids by formation of intermediate acylphosphonium intermediate.

Determination of the geometry of the C=N bond of compound 24 is needed.

Additional NOESY experiments confirmed the E- configuration of >C=N- double bond in molecule of compound 25 (former 24, the numbering was changed).

The English in this manuscript should be revised and edited by a native speaker.

The English was revised and edited.

Line 27: compared –> compared to

compared to

Line 63: 1 –> (1)

(1)

Line 64: the same initial lactone –> sclareolide (1),

sclareolide (1)

Line 64: delete “others intermediate”

Deleted

Line 65: delete “separate”

Deleted

Line 68: was –> were

Were

Line 71: depicted –> in the yields as depicted in the Scheme 1.

in the yields as depicted in the Scheme 1

Line 77: was –> were

The formation.......were proved,

Line 77: signal –> signals

Signals

Line 82: spectra pattern –> spectral data

spectral data

Line 83: instead –> in addition to

in addition to

Line 85: obtained –> afforded

Afforded

Line 86: spectra –> spectrum

Spectrum

Line 87: initial –> starting

Starting

Line 91: base –> basis

Basis

Line 95: at –> according to

according to

Line 105, 114: mechanism –> Reaction pathway

reaction pathway

Line 118: 1 –> (1)

(1)

Line 123: contains –> involve

Involve

Line 162: signal –> signals

Signals

Line 164: of compound 23 –> –> of the structure of compound 24 by

of the structure of compound 25 by (former 24, the numbering was changed)

Line 173: in –> under

Under

Line 177: confirm –> confirmed

Confirmed

Line 179: analysis –> structural analysis

structural analysis

Line 179: of –> by

By

Line 202: mechanism –> reaction pathway

reaction pathway

Line 204: first of all –> initial

Initial

Line 205: delete “and led to obtained before amide”

Deleted

Line 213: reveal –> revealed

Revealed

Lines 214, 216, 217, 220: posses –> possessed

Possessed

Line 221: are –> were

Were

Line 244: remain –> residue

Residue

Line 244: acidulated –> acidified

Acidified

Line 313: additionally –> subsequently

Subsequently

Line 315: delete “obtained in situ”

Deleted

Line 316: delete “dissolved”

Deleted

Reviewer 2 Report

The paper reports the synthesis of new sesquiterpenes for antimicrobial applications.

In general, the manuscript is relevant for the field. However, in the introduction section, the authors need to introduce some applications of the sesquiterpenoids for a better compensation.

On the results, the authors should explain in more detail the interrelation between the stereochemistry of the synthetized compounds and the antifungal or antimicrobial activities.  

They must include the equipment specification.

Some written mistakes should be corrected.

Line 116 Add a coma after was prepared

Line 123 Add a coma after to NMR spectra

Line 123 Verify the verb form The hybrids contains,

Line 163 Verify the verb form  where some signal are

For example, In lines 103, 172 & 190 CHANGE compound by compound.  In line 149 CAHNGE unexpeted BY unexpected. Line 185 CANGE abcence BY absence. 

Verify some terms that may be miswritten such as sustraction, concd, homodrimanic, dublet

Author Response

Comments/suggestions

Answers:

In general, the manuscript is relevant for the field. However, in the introduction section, the authors need to introduce some applications of the sesquiterpenoids for a better compensation.

We agree with you, but the applications of homodrimanic sesquiterpenoids and their biological activities are described well enough in references [10-16] cited in this manuscript, and to a lesser extent in references [17-20].

On the results, the authors should explain in more detail the interrelation between the stereochemistry of the synthetized compounds and the antifungal or antimicrobial activities.

It was mentioned in Results.

They must include the equipment specification.

The specifications were included.

Line 116 Add a coma after was prepared

Added

Line 123 Add a coma after to NMR spectra

Added

Line 123 Verify the verb form The hybrids contains,

The hybrids involve,

Line 163 Verify the verb form where some signal are

some signals are

lines 103, 172 & 190 CHANGE compund by compound.

Compound

line 149 CAHNGE unexpeted BY unexpected

Unexpected

Line 185 CANGE abcence BY absence

Absence

Verify some terms that may be miswritten such as sustraction, concd, homodrimanic, dublet

The terms were verified.

Round 2

Reviewer 1 Report

Following corrections in English are suggested.

All compound numbers should be boldface.

Double arrow in Scheme 5 is not suitable to indicate tautomerization, because the arrow indicates resonance.

In the designation of temperature in Schemes, place blank space between a figure and °C. The circle in °C looks like 0.. 

In the designation of optical rotation in the experimental section, D should not be italic. 

L20: chlorides –> chlorides,

L21: was –> were

L-26: results ­> result

L-28: established –> proposed

L-29: Caspofungin –> caspofungin

L30: Kanamycin –> kanamycin

L28: delete “a”. “Activity” is an uncountable noun.

L31: ditto

L43: ditto

L33: activity –> activities

L63: delete “as promoter”. Lawssons’ reagent is a reagent, not a promoter. It actually reacts.

L-79: first –> at first

L-80: based on –> from

L-80: remove “first”

L82: delete “of the last”

L107: The structures of intermediary compounds as well as final products were

L121: confirm –> confirmed

L-125: 135°, 45°

L-129: becames –> was

L-140: leaded –> led

L-142: triphenylphosphoxide –> triphenylphosphine oxide

L144: leads ­–> led

L145: gives –> gave

L-183: was –> were

L195: delete “a”

L-282: led ­–> lead

L-284: set –> elucidated

L-233: An effort was made –> effort was devoted, “Effort” is an uncountable word.

L-234: delete “mentioned before”

L-278: there were well visible –> down field shifts were observed

L-282: led –> lead

L-330: became –> was, at –> by

L-411: According to the first one, it interacted –> The first role is to interact with axial-----

L411: from amide –> of the amide 

L-429: Caspofungin –> caspofungin

L-433: Kanamycin –> kanamycin

L-500: The –> the

L-524: “Pressure” is an uncountable noun.

L-799: (9 mL) ­–> (9 mL),, by ––> with

L-804: remove second “was added”.

Author Response

Hello!

Thank you very much for suggestion and corrections, for the time and effort invested.

We did all the required in the second revision changes.

Best wishes.

Sincerely,

Prof., Dr. Aculina Aricu
